# A Review of Bioelectrochemical Strategies for Enhanced Polyhydroxyalkanoate Production

**DOI:** 10.3390/bioengineering12060616

**Published:** 2025-06-05

**Authors:** Alejandro Chamizo-Ampudia, Raúl. M. Alonso, Luisa Ariza-Carmona, África Sanchiz, María Isabel San-Martín

**Affiliations:** 1Biochemistry, Department of Molecular Biology, University of León, 24071 León, Spain; asancg@unileon.es; 2Instituto de Biología Molecular, Genómica y Proteómica (INBIOMIC), Universidad de León, Campus de Vegazana, 24071 León, Spain; 3Department of Electrical Engineering and Automatic Systems, University of León, 24071 León, Spain; ralog@unileon.es; 4Faculty of Education and Humanities, International University of La Rioja, 26006 Logroño, Spain; luisamaria.ariza@unir.net

**Keywords:** bioplastics, polyhydroxyalkanoates, sustainable production, waste valorization, PHA biosynthesis, bioelectrochemical systems, microbial electrosynthesis

## Abstract

The growing demand for sustainable bioplastics has driven research toward more efficient and cost-effective methods of producing polyhydroxyalkanoates (PHAs). Among the emerging strategies, bioelectrochemical technologies have been identified as a promising approach to enhance PHA production by supplying electrons to microorganisms either directly or indirectly. This review provides an overview of recent advancements in bioelectrochemical PHA synthesis, highlighting the advantages of this method, including increased production rates, the ability to utilize a wide range of substrates (including industrial and agricultural waste), and the potential for process integration with existing systems. Various bioelectrochemical systems (BES), electrode materials, and microbial strategies used for PHA biosynthesis are discussed, with a focus on the roles of electrode potentials and microbial electron transfer mechanisms in improving the polymer yield. The integration of BES into PHA production processes has been shown to reduce costs, enhance productivity, and support the use of renewable carbon sources. However, challenges remain, such as optimizing reactor design, scaling up processes, and improving the electron transfer efficiency. This review emphasizes the advancement of bioelectrochemical technologies combined with the use of agro-industrial waste as a carbon source, aiming to maximize the efficiency and sustainability of PHA production for large-scale industrial applications.

## 1. Introduction

Bioelectrochemical technologies have emerged as innovative and sustainable platforms to enhance the microbial production of polyhydroxyalkanoates (PHAs), a family of biodegradable and biocompatible polyesters that are regarded as promising alternatives to conventional petroleum-based plastics. Bioelectrochemical systems (BESs) have been increasingly employed in bioprocess engineering due to their ability to integrate microbial metabolism with electrochemical principles [1,2,3,4]. In these systems, electroactive microorganisms are utilized to catalyze redox reactions in the presence of an electrode, allowing the conversion of chemical energy derived from organic substrates into electrical energy and vice versa [5]. This unique capacity to manipulate electron flows has been harnessed not only to generate electricity but also to stimulate and optimize the microbial pathways involved in the biosynthesis of value-added products, including PHAs, under controlled electrochemical conditions [6,7]. One of the most promising configurations of BESs for PHA production involves the use of dual-chamber microbial fuel cells (MFCs), in which substrate oxidation by microorganisms in the anodic chamber is coupled with the generation of an electrical current and the simultaneous conversion of carbon sources into intracellular PHA granules [8]. This dual functionality allows for improved energy efficiency and resource utilization, while contributing to the overall sustainability and circularity of biorefinery processes [9]. The range of raw materials suitable for PHA production is remarkably broad, encompassing simple carbohydrates such as glucose and sucrose, vegetable oils, and glycerol—a major byproduct of biodiesel production—thus enabling the valorization of low-cost and renewable feedstocks [3]. Furthermore, the use of industrial and municipal wastewater as a carbon source has gained increasing attention as an economically and environmentally attractive option, offering the potential to combine waste treatment with biopolymer synthesis [3]. In parallel, advances in synthetic biology and plant biotechnology have led to the development of transgenic crops engineered to express bacterial PHA biosynthetic genes, which allows the direct accumulation of PHAs within plant tissue [10]. This approach represents a significant step toward integrating agricultural production with industrial biotechnology, potentially reducing the production costs and expanding the scope of PHA manufacturing systems [11]. Several companies and research initiatives are actively investigating these strategies, including the use of BESs and unconventional feedstocks, as part of a broader effort to promote sustainable materials and reduce the dependence on fossil-based plastics. Notable contributors to the development of PHA production from waste streams and bioelectrochemical processes include Veolia subsidiary Anoxkaldnes, as well as start-ups such as Micromidas, Mango Materials, Full Cycle Bioplastics, Newlight, and Paques Biomaterials [12]. Once produced, PHAs exhibit excellent versatility in their downstream processing, as they can be shaped into films, fibers, and molded articles using standard techniques such as injection molding and extrusion [13]. This compatibility with existing polymer-processing technologies facilitates the integration of PHAs into commercial applications across diverse sectors, including packaging, agriculture, and biomedicine, and underscores their potential as scalable and sustainable bioplastics for a future bio-based economy [14].

## 2. Polyhydroxyalkanoates (PHAs): Characteristics and Production

### 2.1. Types of PHA

Polyhydroxyalkanoates (PHAs) represent a diverse group of biopolyesters synthesized by a wide range of microorganisms, serving as intracellular storage materials for carbon and energy, typically under conditions where nutrients are limited but carbon sources are abundant [3,15]. These biopolymers have garnered significant attention in recent years due to their biodegradability, renewability, and the ability to tailor their material properties, positioning them as promising substitutes for conventional plastics in numerous industrial applications [16]. PHAs are composed of hydroxyalkanoate monomers linked through ester bonds, and the properties of these polymers can be manipulated by varying the composition of the monomers, their chain lengths, and the molecular weight of the polymer [3].

PHAs can be broadly classified into two main categories based on the lengths of the monomers’ carbon chains: short-chain-length PHAs (scl-PHAs) and medium-chain-length PHAs (mcl-PHAs) [3] (Figure 1). scl-PHAs consist of monomers containing three to five carbon atoms, while mcl-PHAs are composed of monomers with carbon chains ranging from six to fourteen atoms [3]. Among scl-PHAs, poly(3-hydroxybutyrate) (PHB) is the most extensively studied and commercially produced homopolymer [17]. PHB is known for its high crystallinity and tensile strength, which lends it good mechanical properties in static applications [18]. However, PHB suffers from inherent brittleness and poor elongation at break, which limits its use in dynamic and impact-resistant applications [19]. These shortcomings have led to the development of various copolymers, such as poly(3-hydroxybutyrate-co-3-hydroxyvalerate) (PHB-co-HV), which incorporates 3-hydroxyvalerate (HV) monomers [19,20]. Adding HV reduces the polymer’s crystallinity, thus enhancing its flexibility, toughness, and overall processability [21]. The ratio of hydroxybutyrate (HB) to HV in the copolymer can be adjusted, allowing for the tuning of the mechanical and thermal properties [21]. For example, higher HV content improves the elasticity and lowers the polymer’s melting temperature, making it more suitable for a wider array of applications, including packaging and biomedical uses [21].

Another significant copolymer is poly(3-hydroxybutyrate-co-3-hydroxyhexanoate) (PHB-co-HHx), which incorporates 3-hydroxyhexanoate (HHx) monomers [23]. This copolymer exhibits even greater ductility and reduced crystallinity compared to PHB-co-HV, enhancing its suitability for applications in tissue engineering, drug delivery, and other biomedical fields [24]. The incorporation of HHx units further contributes to the material’s improved flexibility, lower melting temperature, and better processability, making it an attractive material for applications requiring biocompatibility and biodegradability [24].

In contrast to scl-PHAs, mcl-PHAs, typically produced by microorganisms such as Pseudomonas species, exhibit elastomeric properties and lower crystallinity and melting points [3,25]. These mcl-PHAs are characterized by their high flexibility, extensibility, and compatibility with hydrophobic compounds [26]. Such properties make mcl-PHAs ideal candidates for use in applications where flexibility and stretchability are critical, such as in medical adhesives, wound dressings, coatings, and cosmetic formulations [27]. The synthesis of mcl-PHAs is often accomplished by utilizing fatty acids or other hydrophobic substrates, which provides an avenue for the use of waste oils and other low-cost materials as feedstocks, further enhancing the sustainability of their production [3].

The diversity of PHAs is further expanded by incorporating additional monomers into the polymer backbone [15]. For example, poly(3-hydroxybutyrate-co-4-hydroxybutyrate) (P3HB-co-4HB) is a copolymer that incorporates 4-hydroxybutyrate (4HB), which significantly enhances the polymer’s elasticity, making it particularly well suited for biomedical applications such as tissue scaffolds, controlled drug release systems, and surgical sutures [28]. The flexibility of PHAs means that they can be further tuned by adjusting the ratio of 3HB to 4HB, enabling the production of materials with specific mechanical and degradation characteristics [28,29].

Recent advances in metabolic engineering and synthetic biology have enabled the production of tailor-made PHAs with novel monomers, imparting unique physicochemical properties [30]. These engineered PHAs may exhibit enhanced thermal resistance, UV stability, improved hydrophilicity, or other desirable attributes that are tailored to specific applications [31]. The ability to manipulate the monomer composition through the optimization of the microbial strains, fermentation conditions, and substrate selection provides a high degree of flexibility in producing PHAs with application-specific properties [32].

This structural versatility, combined with the biodegradability and environmental compatibility of PHAs, makes them suitable for a wide range of applications [33]. These include biodegradable packaging materials, agricultural films, medical implants, drug delivery systems, and other disposable products [33]. Moreover, the potential to produce PHAs from industrial byproducts, organic waste, or CO_2_-derived substrates enhances the sustainability and economic feasibility of their production [3]. As a result, PHAs are considered a promising alternative for the development of next-generation bioplastics that meet both performance and environmental sustainability criteria.

Additionally, the integration of PHAs into emerging biotechnologies, such as bioelectrochemical systems and synthetic biology, has opened up new avenues by which to enhance their production efficiency [7]. Bioelectrochemical systems, for instance, allow for the conversion of renewable resources and waste into PHAs, aligning with the principles of a circular bioeconomy [3,7,33,34]. The development of new microbial strains and fermentation processes, coupled with advances in process optimization, offers the potential to scale up PHA production, making it a viable option for large-scale industrial applications.

PHAs represent a highly promising class of bioplastics that not only offer significant advantages in terms of environmental sustainability but also provide a versatile platform for the creation of materials with tailored properties. Their ongoing development, driven by advances in biotechnology and synthetic biology, is expected to play a crucial role in the transition to a bio-based economy [7,35]

### 2.2. Microorganisms Involved in PHA Production

A wide array of bacterial species have been reported to synthesize and accumulate polyhydroxyalkanoates (PHAs) as intracellular energy and carbon reserves, particularly under unbalanced nutritional conditions, such as limitations in essential nutrients like nitrogen, phosphorus, or oxygen, in the presence of excess carbon [3,32]. These biopolymers are stored in the form of granules within the cytoplasm and serve as metabolic buffers under stress conditions [36]. Among the most prominent and industrially relevant PHA-producing microorganisms is *Cupriavidus necator* (formerly *Ralstonia eutropha*), a Gram-negative β-proteobacterium capable of accumulating up to 90% of its dry cell weight as PHB under optimized conditions [3]. This strain is widely regarded as the model organism for PHA biosynthesis due to its robust growth, high polymer yields, and well-characterized metabolic pathways. It is known to utilize a broad range of carbon substrates, including glucose, sucrose, glycerol, fatty acids, and even organic acids derived from waste streams, making it highly adaptable to industrial processes [37]. *Pseudomonas putida*, another key PHA-producing bacterium, is notable for its ability to synthesize medium-chain-length PHAs (mcl-PHAs), which are elastomeric, less crystalline, and more flexible than short-chain-length PHAs (scl-PHAs) [25,37,38]. This species can metabolize a variety of hydrophobic substrates, including long-chain fatty acids, alkanes, and aromatic compounds, and is frequently employed in the biosynthesis of tailor-made copolymers for applications in packaging, biomedicine, and soft materials [39,40]. *Azotobacter vinelandii*, a nitrogen-fixing soil bacterium, is capable of PHA production under fully aerobic conditions, often without the need for nutrient limitations, which simplifies its cultivation and reduces the operational costs [41]. It has been shown to produce both PHB and PHB-co-HV depending on the carbon source supplied [42]. *Bacillus megaterium*, a Gram-positive bacterium with generally regarded as safe (GRAS) status, has gained attention due to its ability to grow on inexpensive substrates and to form PHAs without stringent sterile conditions [43]. Its thick cell walls and spore-forming capabilities enhance its resilience under industrial fermentation environments [44]. *Alcaligenes latus* is another fast-growing PHA producer, known for its ability to accumulate PHAs rapidly in batch and fed-batch systems, with minimal nutrient limitations and relatively short fermentation cycles, which is advantageous for continuous or semi-continuous PHA production [45]. The use of wild-type strains is frequently complemented by metabolic and genetic engineering strategies to enhance the monomer diversity, increase the carbon flux toward PHA biosynthesis, and improve the tolerance to precursor toxicity [46]. Novel extremophilic and marine bacteria, including halophiles such as *Halomonas* spp., have also been investigated due to their ability to grow under high-salinity conditions, reducing the need for strict sterilization and freshwater input [47].

In addition to these bacterial systems, various microalgae and cyanobacteria have also been explored for PHA production, offering the significant advantage of utilizing carbon dioxide as the primary carbon source through photosynthetic processes. Species such as *Synechocystis* sp. PCC 6803, *Spirulina platensis*, and *Chlorella vulgaris* have demonstrated the ability to accumulate PHAs under nutrient-limited and stress-induced conditions [48,49]. These photoautotrophic microorganisms leverage the Calvin–Benson–Bassham (CBB) cycle, driven by light energy, to fix atmospheric CO_2_ into organic intermediates, which can subsequently be channeled toward PHA biosynthesis. Although the PHA yields from microalgae and cyanobacteria are typically lower compared to heterotrophic bacteria, their low nutrient requirements, potential integration with wastewater treatment systems, and direct utilization of CO_2_ make them attractive candidates for sustainable and eco-friendly bioplastic production [48,50]. Furthermore, ongoing efforts in metabolic engineering and cultivation optimization aim to enhance the carbon flux toward PHA accumulation in these photosynthetic platforms, broadening the scope for large-scale, carbon-neutral PHA production.

This diversity of microbial hosts, each with unique physiological and metabolic traits, enables the production of a wide spectrum of PHAs with varying mechanical, thermal, and degradation properties, thereby expanding their potential for commercialization in a range of sectors, including agriculture, food packaging, biomedicine, and environmental remediation. In addition, among these microorganisms, there are promising candidates with potential applications in bioelectrochemical processes, where their metabolic versatility and electron transfer capabilities could be harnessed for sustainable energy production, bioremediation, and bioplastic synthesis (Table 1).

### 2.3. Growth Conditions and Carbon Sources

The production of polyhydroxyalkanoates (PHAs) is influenced by a range of factors that significantly impact both the efficiency and the quantity of the polymer produced [67]. Among the most critical factors are the carbon sources used, the growth conditions in which the microorganisms are cultured, and the genetic modifications of the microbial strains [3]. Each of these factors plays an essential role in optimizing the PHA production process, enabling higher yields and more sustainable bioplastic production [68,69].

Carbon sources are crucial in the production of PHAs because they provide the necessary building blocks for polymer synthesis and the energy required by microorganisms for growth [3]. In addition to being a source of carbon, they also serve as electron donors in microbial metabolism, playing an essential role in the electron transport chain, which is responsible for generating the ATP needed for PHA biosynthesis [70]. Traditionally, simple sugars such as glucose and sucrose have been the preferred carbon sources for PHA production [3]. These sugars are easily metabolized by most PHA-producing bacteria, which can rapidly grow and accumulate PHAs under favorable conditions [3]. However, the use of these traditional carbon sources can be expensive, particularly in large-scale production [71]. As a result, there has been growing interest in utilizing alternative and more sustainable carbon sources, particularly industrial and agricultural waste, which can be abundant, inexpensive, and renewable [72].

Agroindustrial residues, such as corn stover, sugarcane molasses, wheat straw, and fruit peels, have been identified as viable carbon sources for PHA production [3]. These residues are often rich in carbohydrates, fatty acids, and other organic compounds, which can be converted into fermentable sugars or volatile fatty acids (VFAs) through pretreatment processes [3,72]. Once these components are released, they can serve as excellent carbon sources for microbial fermentation [3,73]. Using waste materials such as these not only reduces the reliance on fossil-based raw materials but also helps to manage waste streams that would otherwise be discarded, contributing to a more sustainable and circular economy [74,75,76]. Studies have shown that PHA production from agricultural and industrial residues is not only feasible but can also lower the overall cost of production, especially when compared to the use of pure sugars [73,77,78].

Moreover, industrial waste streams, including byproducts from the production of biodiesel or food processing, have also proven to be effective carbon sources [3,79]. These waste products, rich in organic carbon, can be pretreated to release simpler sugars or fatty acids, which microorganisms can readily consume to produce PHAs [3,32]. The use of such waste streams has the dual benefit of reducing the environmental impact of industrial processes while also providing an affordable and renewable source of carbon for bioplastic production [80].

Alongside the selection of carbon sources, the growth conditions in which the microorganisms are cultivated are another critical factor that influences the yield and accumulation of PHAs [81]. Oxygen availability is one of the most important conditions affecting PHA production [82]. In many bacterial species, oxygen limitation is a key trigger for PHA accumulation [32,83,84,85]. Under conditions of oxygen scarcity, microorganisms divert the available carbon toward the synthesis of PHAs as a storage material for energy [32]. Managing the oxygen levels can, therefore, be a powerful tool by which to enhance PHA production [86]. Similarly, controlling the pH of the fermentation medium is vital to ensure optimal growth and polymer production [32]. Each bacterial strain has an ideal pH range in which it thrives—generally between 6.5 and 8.0—and deviations from this range can lead to stress, which negatively impacts PHA synthesis [87]. The temperature also plays a key role, with each microorganism having an optimal growth temperature that should be maintained throughout the fermentation process [32]. Nutrient limitations, such as nitrogen or phosphorus limitations, are often employed to promote PHA accumulation by limiting cell growth and directing more carbon to polymer production instead [3,32].

Genetic modifications of microbial strains can significantly enhance the production of PHAs by improving the ability of microorganisms to produce higher yields or to use a broader range of carbon sources [3,88]. Through genetic engineering, the metabolic pathways involved in PHA synthesis can be optimized to improve the efficiency of polymer production [89]. For example, overexpressing key genes in the PHA biosynthesis pathways or introducing new metabolic routes can boost PHA accumulation in microbial cells [90]. In *Cupriavidus necator*, the well-characterized *phaCAB* operon is known to encode β-ketothiolase (*phaA*), acetoacetyl-CoA reductase (*phaB*), and PHA synthase (*phaC*), which are considered essential for the conversion of acetyl-CoA into polyhydroxybutyrate (PHB) [91]. Recently, metabolic modeling and genetic variability have been explored to optimize PHA production in this organism. In *Pseudomonas putida*, which is responsible for the biosynthesis of medium-chain-length PHAs (mcl-PHAs), various engineering strategies have been implemented. These have included the overexpression of genes such as *phaC1*, *phaC2*, *phaG*, and *alkK*, and the deletion of genes including *phaZ* (PHA depolymerase) and elements of the β-oxidation pathway (*fadBA1*, *fadBA2*), in order to improve precursor availability and increase polymer accumulation [39,92]. Moreover, *phaC2* has been engineered—guided by structural analyses—to eliminate substrate trapping, resulting in substantial improvements in mcl-PHA production from substrates such as crude glycerol [93]. In addition, PHAs with tailored properties, such as enhanced thermal stability, flexibility, or biodegradability, can be obtained [94]. These engineered strains are often more robust and can achieve higher yields compared to wild-type strains, making the production process more cost-effective and scalable [95].

One promising development in improving the PHA production efficiency is the integration of electrochemical methods into the fermentation process [89]. Electrochemical approaches, which involve the use of electrodes to provide an external electron source, can significantly enhance PHA production by improving the efficiency of the electron transport chain [96,97]. By providing a supplementary source of electrons, the electrochemical process reduces the need for microorganisms to allocate carbon sources to energy production, allowing more carbon to be diverted toward PHA biosynthesis instead [98]. This can result in higher yields of PHAs without the need for additional carbon sources, thus improving the overall production efficiency [99]. Additionally, electrochemical methods can support the microbial reduction of carbon dioxide (CO_2_), potentially transforming a greenhouse gas into a valuable resource for PHA production, further contributing to a sustainable process [100].

In summary, the efficient production of PHAs relies on the careful selection of carbon sources, optimal growth conditions, and genetic modifications of microbial strains. The use of agroindustrial residues and industrial waste streams as carbon sources has gained considerable attention due to its cost-effectiveness, sustainability, and ability to contribute to waste management. Furthermore, the incorporation of electrochemical methods into the fermentation process can further enhance PHA production by improving the efficiency of the electron transport chain and enabling more carbon to be directed toward polymer synthesis. Together, these strategies can significantly increase the yield and sustainability of PHA production, making it a viable alternative to conventional plastics in a wide range of applications.

## 3. Bioelectrochemistry Applied to PHA Production

### 3.1. Principles of Bioelectrochemical Systems (BES)

Bioelectrochemical systems are hybrid platforms that integrate microbiological catalysis with electrochemical systems, offering a sustainable interface for both energy production and chemical synthesis (Figure 2). While originally developed for electricity generation in microbial fuel cells (MFCs) [101], BESs have increasingly gained attention for their potential in electrosynthesis—the microbial-driven formation of value-added chemicals using electrical energy and low-cost substrates like CO_2_, wastewater, or organic residues.

At the core of BES-based electrosynthesis is the capacity of electroactive microorganisms to perform extracellular electron transfer (EET). These microbes can either donate electrons to an anode or receive electrons from a cathode, thus catalyzing the reduction or oxidation reactions depending on the system configuration. In microbial electrosynthesis (MES), electrons supplied by the cathode are used by microbes to reduce carbon sources (e.g., CO_2_) into compounds such as acetate, butyrate, or even more complex organics [102].

A typical BES for electrosynthesis comprises a biofilm-covered electrode immersed in a chamber containing a suitable electron donor or acceptor. In MES systems, CO_2_ reduction occurs at the cathode or in the bulk, where microbes utilize electrons delivered from the electrode to drive endergonic biosynthetic pathways, leading to the production of platform chemicals [103]. Notably, these microbial electrosynthesis processes can be tailored to enable chain elongation [104], whereby short-chain carbon compounds like acetate are biologically upgraded into medium-chain carboxylates through sequential reduction and condensation reactions. These reactions offer a biologically mild, low-energy alternative to conventional thermochemical synthesis.

Recent research has shown that combining BESs with synthetic biology and metabolic engineering can enhance product selectivity and increase yields by modifying the microbial chassis to produce specific compounds of interest [105,106]. Thus, BESs for electrosynthesis represent a promising solution for decentralized chemical manufacturing, enabling the transformation of waste streams and renewable electricity into valuable biochemicals. However, challenges remain in terms of product recovery, process efficiency, and economic feasibility, which must be addressed before these systems can be industrially deployed.

### 3.2. Reactor Configurations and Limitations in Microbial Electrosynthesis for Value-Added Chemicals

MES is a promising bioelectrochemical platform that could utilize renewable electricity to drive the microbial reduction of CO_2_ into value-added chemicals such as acetate, methane, or other complex compounds. The performance of MES systems, however, is tightly linked to the reactor configuration, the availability of electron donors and CO_2_, and the physicochemical limitations of the system.

Recent progress in reactor engineering has significantly enhanced MES productivity. A compact plate reactor design with a zero-gap anode and large surface area with reticulated vitreous carbon cathodes demonstrated acetate and methane production rates that were competitive with those of conventional fermentation processes. For example, the MES of acetate by *Thermoanaerobacter kivui* reached up to 3.5 g/L·h at titers of 14 g/L, approaching the performance of glucose-fed chemostats and continuous gas fermentation systems [107]. Similarly, the methane production rates when using mixed cultures exceeded 10 L/L·d, highlighting the efficacy of the reactor architecture in overcoming mass transfer and current density constraints [108].

Despite these advances, MES systems are still limited by the availability of CO_2_, particularly in configurations that rely on low-flow or diluted gas streams. Unlike glucose or acetate fermentation, MES systems are highly dependent on the continuous, efficient delivery of CO_2_ to the cathodic biofilm. Inadequate CO_2_ partial pressures or suboptimal gas transfer can bottleneck microbial metabolism. For instance, a comparative analysis has shown that CO_2_ delivery was more efficient when switching from N_2_/CO_2_ (80/20) mixtures to pure CO_2_, resulting in improved acetate productivity [107]. Furthermore, insufficient gas tightness or diffusive CO_2_ losses across membranes also diminish the reactor performance, especially at low current densities [108].

This limitation is consistent with the broader challenges reported in anaerobic gas fermentation. The low solubility of CO_2_ and H_2_ in aqueous media severely constrains the rate of microbial conversion. Ale Enriquez and Ahring (2023) [109] emphasized that increasing the CO_2_ partial pressure—through pressurization or using membrane contactors—can significantly enhance the gas fermentation yields. They also noted that industrial CO_2_-rich streams like biogas and flue gas are attractive yet challenging feedstocks due to their variability and dilution.

To mitigate the CO_2_ limitations in MES, several strategies are under consideration: (i) increasing gas–liquid mass transfer through pressure control, (ii) optimizing the cathode porosity and flow-through designs to improve the CO_2_ accessibility at the biofilm surface, and (iii) employing reactor architectures that minimize the headspace and reduce diffusion losses. For example, the latest plate reactor designs achieve higher acetate productivity not only due to optimized hydrogen delivery but also due to improved flow dynamics, enabling them to maintain CO_2_ availability even under a high microbial demand [107].

## 4. Enhancing PHA Production Through Bioelectrochemical Technologies

By integrating biological processes with electrochemical techniques, BESs offer unique advantages for the control and optimization of PHA biosynthesis. This approach leverages the ability of microorganisms to utilize electrons derived from electrodes, thereby influencing metabolic pathways and improving the PHA yields. Research has demonstrated the potential of BESs to enhance PHA production by manipulating electron flows and creating favorable conditions for microbial growth and PHA accumulation, using organic matter in waste streams as the substrate for anodic microorganisms [110,111].

### 4.1. Metabolic Optimization via Electrical Stimulation

The application of an external voltage in BESs serves as a powerful tool to regulate metabolic pathways [112] that are crucial for PHA biosynthesis. Electrical stimulation allows for precise control over electron transfer to microorganisms, leading to increased reducing power [113], enhanced precursor availability [114], and the modulated transcription of key PHA biosynthetic genes. Studies have shown that, by optimizing the applied voltage, electrode materials, and reactor configurations, it is possible to significantly improve the PHA production rates and yields [4]. Moreover, electrical stimulation can influence the metabolic flux toward PHA synthesis, enhancing the overall efficiency of the process [115]. This approach offers a promising avenue for sustainable PHA production by utilizing renewable energy sources and minimizing the environmental impact.

### 4.2. Alternative Carbon Sources and Their Bioelectrochemical Conversion

The economic viability and sustainability of polyhydroxyalkanoate (PHA) production are strongly influenced by the cost and availability of carbon feedstocks [116]. Traditional PHA production relies heavily on refined sugars and vegetable oils, which can be expensive and prevent PHAs from being cost-effective alternatives to petroleum-derived plastics [77]. To address these limitations, bioelectrochemical systems are being explored to valorize alternative, low-cost carbon sources for PHA synthesis [8]. This approach not only reduces the production costs but also contributes to waste reduction and resource recovery, aligning with the principles of a circular economy.

#### 4.2.1. Agricultural Residues

Lignocellulosic hydrolysates derived from agricultural residues represent a promising feedstock for BES-driven PHA production [117]. These residues, such as rice husk, apple, grape, corn stover, banana peel, and walnut shells, are abundant and readily available [118]. Through pretreatment and enzymatic hydrolysis, lignocellulosic biomass can be converted into a mixture of sugars, which can then be utilized by electroactive microorganisms in BESs [119] to produce PHA precursors. Indeed, research has found that an applied potential of 150 mV, utilizing a reactor equipped with two graphite electrodes (10 cm in length and 0.6 cm in diameter), significantly enhances the surface morphology and functionality of hemicellulose, leading to improved substrate uptake capacities under defined solid load and alkaline conditions, thereby favoring efficient substrate utilization for product recovery.

#### 4.2.2. Industrial Byproducts

Industrial byproducts, including glycerol, wastewater, and food processing residues, offer another avenue for cost-effective PHA production using BESs. Glycerol, a byproduct of biodiesel production, can be efficiently converted into PHA [120] and it is furthermore extensively utilized as a substrate within microbial electrochemical systems [121,122]. For instance, one study effectively demonstrated this conversion in a 0.4-L reactor equipped with graphite plate electrodes (5 × 20 cm), applying a potential scan rate of 0.1 mV s^−1^ [121]. Concurrently, other investigations have successfully employed larger, specialized reactor designs, such as a lamellar-type configuration featuring multiple anode-cathode compartments (e.g., each cathode frame with an inner volume of 0.56 L housing three IGS-743 graphite electrodes of 4 × 30 × 150 mm dimensions) to optimize the use of glycerol and other byproducts for PHA synthesis [122]. Similarly, wastewater streams containing organic pollutants and food processing residues rich in carbohydrates and fatty acids can be valorized as substrates for PHA synthesis [123]. This approach not only reduces waste disposal costs but also transforms these byproducts into valuable biopolymers.

#### 4.2.3. CO_2_ Fixation

Bioelectrochemical systems that utilize CO_2_ fixation represent a particularly innovative and sustainable strategy for PHA production. In these systems, microorganisms capable of autotrophic growth utilize electrons provided by electrodes to reduce CO_2_ to organic molecules, which serve as precursors for PHA biosynthesis [124,125]. This approach offers the potential to produce PHAs from a greenhouse gas, contributing to carbon capture and utilization strategies. Recent studies have demonstrated the feasibility of this process; for instance, investigations employing indium nanoparticle electrodes as the working electrode (with a surface area of 1.7 cm^−2^) have shown promising results in CO_2_ conversion [124]. Furthermore, the microbial-catalyzed cathodic conversion of CO_2_ into key intermediates like acetate has been reported at a relatively low cathode potential—approximately −0.4 V—versus a standard hydrogen electrode (SHE), highlighting the energy efficiency of this route [126]. While still under development, CO_2_-based PHA production holds significant promise for a future bioeconomy.

### 4.3. Electrodes as Electron Acceptors and Donors

In BESs designed for PHA production, electrodes serve as pivotal components, acting as both electron acceptors and donors to manipulate microbial metabolism. When functioning as electron acceptors, electrodes enable microorganisms to oxidize organic substrates, effectively driving intracellular carbon flux toward the synthesis of PHA precursors such as volatile fatty acids [125,127]. This process is crucial in enhancing the availability of acetyl-CoA and other essential building blocks, thereby promoting efficient PHA accumulation [128]. By carefully controlling the electrode potential, researchers can optimize the oxidation of specific substrates and steer metabolic pathways toward PHA biosynthesis [7].

Conversely, electrodes also function as electron donors, supplying microorganisms with the electrons necessary for reduction reactions that support microbial growth and the polymerization of PHA monomers [100]. This reduction process is vital in maintaining cellular redox balance and providing the energy required for polymer synthesis. The ability to precisely regulate electron donation through electrode manipulation allows for the optimization of the microbial growth conditions and the control of the PHA polymer chain length and composition [129]. Therefore, the dual functionality of electrodes in BESs is essential in maximizing the PHA production efficiency and tailoring the polymer characteristics to specific applications.

## 5. State of the Art and Recent Advances

The integration of BESs into bioplastic production pathways has evolved significantly in recent years, transitioning from proof-of-concept studies to more complex and efficient setups. Early work demonstrated the potential of microaerophilic environments at the cathode to modulate microbial metabolism and promote PHA accumulation. In one such study, utilizing a system equipped with non-catalyzed graphite plate electrodes (a plain cathode with a surface area of 70 cm^2^ and a perforated anode with 83.5 cm^2^), the creation of a low-oxygen microenvironment favored electrogenesis and directed metabolic flux toward PHA biosynthesis, resulting in PHA content of 19% of the dry cell with a high proportion of 3-hydroxybutyrate (89%) [130]. This approach emphasizes the importance of electron availability and redox conditions in driving PHA production.

Building on this foundation, researchers have explored strategies to enhance electron uptake by biocatalysts. For instance, the use of immobilized redox mediators such as Prussian Blue significantly improved direct electron transfer to *Rhodopseudomonas palustris* TIE-1, leading to a 1.4-fold increase in PHB production compared to uncoated electrodes [131]. This particular study utilized pure graphite rod electrodes (with a surface area of 5.149 cm^2^) as the working electrodes. This approach contrasts earlier findings where soluble iron mediators were ineffective, highlighting the role of the electrode design and surface chemistry in optimizing BES performance.

Further innovations focused on carbon dioxide valorization through MES. A two-stage process combining CO_2_ fixation and VFA production with subsequent feeding to PHA-producing bacteria achieved a notable yield of 0.41 kg of PHA carbon per kg of CO_2_ carbon [132], and 0.460 g/L PHB was produced ex situ from 5 g/L acetate in a triple-chamber MES reactor [133]. In comparison, more recent single-stage systems employing co-cultures of acetogenic and PHB-accumulating bacteria have demonstrated significant improvements in process integration and efficiency. Notably, a symbiotic MES configuration achieved a 7.14-fold increase in the PHB yield under an optimized voltage and electrode functionalization with a poly(3,4-ethylenedioxythiophene)–poly(styrenesulfonate) polymer [100], thus bypassing intermediate extraction steps and offering a streamlined route from CO_2_ to PHB. This particular system employed a stainless-steel (SS) mesh anode (measuring 2 × 10 cm, with an effective surface area of 1 mm^2^) and a graphite plate cathode (2 × 10 cm, with a thickness of 0.5 cm) as the electrodes, demonstrating the impacts of the electrode material and design on process efficiency.

The use of alternative feedstocks has also gained traction. In an MFC setup, *Arenibacter palladensis* was employed to hydrolyze chitin biomass and generate N-acetylglucosamine, which was then utilized by *Ralstonia eutropha* for PHA synthesis. This system simultaneously produced electricity and achieved notable PHA yields, showcasing the feasibility of using marine biomass as a substrate in BESs [134]. Similarly, textile wastewater served as both an electron donor and carbon source in a dual-chamber MFC, enabling simultaneous electricity generation, pollutant removal, and PHA production by Enterobacter spp., with the PHA content reaching over 56% of the cell dry mass [8]. These studies underscore the potential of BESs in coupling waste valorization with biopolymer synthesis.

Recent work has also explored the synergy between MES and microalgae cultivation. One study utilized volatile fatty acids produced from CO_2_ in MES as feedstock for microalgae, achieving substantial astaxanthin accumulation [135]. While this system was designed for antioxidant production, the underlying principle of integrating MES effluents with microbial or algal cultures could be extended to PHA-producing organisms, offering new avenues for the co-production of bioplastics and high-value compounds.

Lastly, photoelectrosynthesis using mixed cultures of purple phototrophic bacteria under heterotrophic conditions was shown to yield higher PHB titers (up to 58.23 mg/L) than previous photoautotrophic approaches using pure cultures [136]. This improvement was linked to the upregulation of PHA biosynthetic enzymes at lower applied voltages, emphasizing how the electron pressure and metabolic regulation are tightly coupled in these systems. Complementarily, a recent study leveraging microbial consortia and adaptive evolution techniques achieved PHB accumulation of up to 73% of the cell volume in *Cupriavidus necator*, suggesting the occurrence of direct interspecies electron transfer in BESs [7], a mechanism that could redefine the electron flow dynamics in these environments.

Together, these studies illustrate the progressive refinement of BES design and microbial engineering for PHA production. From microaerophilic control [130] to redox mediator optimization [131], CO_2_ conversion [100,132], waste valorization [8,134], and phototrophic integration [135,136], the field is moving toward more efficient, scalable, and sustainable bioprocesses. A comparative look at recent strategies reveals a clear trend: systems that integrate metabolic cooperation, optimized electron transfer interfaces, and non-conventional substrates show the greatest promise for industrial-scale application.

## 6. Challenges and Future Perspectives

Despite the significant advances in integrating BESs for PHA production, several scientific and technological challenges remain that limit their large-scale application and commercial viability. One of the most pressing issues is the relatively low production yields and rates compared to conventional fermentation processes. While the optimization of electrode materials and configurations has improved electron transfer and PHA accumulation in some systems [7,100,131], achieving industrially relevant titers with cost-effective setups is still a challenge.

Another major limitation lies in the metabolic efficiency and robustness of the microbial consortia used in BESs. Many electroactive bacteria are not naturally high PHA producers, and, conversely, efficient PHA-accumulating strains often lack extracellular electron transfer capabilities. Although recent approaches have used synthetic co-cultures or symbiotic systems to bridge this gap [7,100], maintaining long-term stability, preventing community shifts, and avoiding contamination are challenges that require further development in microbial ecology, metabolic engineering, and bioprocess control.

The availability and complexity of substrates also pose practical challenges. While BESs allow for the valorization of unconventional and low-cost carbon sources [137]—such as CO_2_ [135], chitin biomass [134], and textile wastewater [138]—these feedstocks often require pretreatment or careful system design to ensure compatibility with microbial metabolism and electrochemical conditions. Moreover, systems using gaseous substrates like CO_2_ must overcome mass transfer limitations and ensure sufficient reduction equivalents through efficient electron transfer mechanisms [139].

From an engineering perspective, electrode materials and reactor design are critical aspects that influence the system performance, scalability, and cost [140]. Although the functionalization of electrodes with graphene oxide, polymers, or redox mediators has shown promise [100,131,141], issues related to long-term stability, fouling, and economic feasibility remain unresolved. Furthermore, most studies to date are limited to laboratory-scale setups, and the transition to the pilot or industrial scale demands rigorous techno-economic analysis and life cycle assessment to validate the environmental benefits and identify economic bottlenecks.

Looking forward, integrating BESs for PHA production within circular bioeconomy frameworks offers a promising pathway for sustainable bioplastic manufacturing. Future research should focus on developing genetically engineered strains capable of coupling extracellular electron uptake with efficient PHA biosynthesis, improving reactor designs for better mass and electron transfer, and exploring hybrid systems that combine BESs with phototrophic or algal processes. Advances in omics technologies and bioinformatics can also aid in elucidating the molecular mechanisms underpinning electron-driven metabolism, enabling more targeted system optimization.

## 7. Conclusions

Bioelectrochemical systems offer a transformative strategy by which to improve the production of PHAs, presenting a synergistic platform that integrates microbial metabolism with a controlled electron flow to enhance biopolymer synthesis. This review has highlighted how electrical stimulation can modulate intracellular redox states, increase the availability of key precursors such as acetyl-CoA, and influence the transcription of biosynthetic genes to steer microbial pathways toward efficient PHA accumulation. Moreover, the dual functionality of electrodes as both electron donors and acceptors enables precise control over the microbial activity, redox balance, and metabolic fluxes.

The integration of BESs with unconventional and low-cost feedstocks—including agricultural residues, industrial byproducts, and even CO_2_—demonstrates the potential of this approach to reduce costs and enhance sustainability, aligning with the goals of a circular bioeconomy. Advances in reactor configurations, electrode materials, and microbial consortia design have significantly improved the PHA yields and selectivity. Innovations such as symbiotic co-cultures, functionalized electrodes, and coupling with microbial electrosynthesis have expanded the range of feasible substrates and increased system productivity.

However, challenges remain regarding their long-term operational stability, electron transfer efficiency, microbial compatibility, and process scalability. Addressing these limitations will require integrated efforts in synthetic biology, bioengineering, and systems optimization. Looking forward, the development of genetically engineered strains capable of coupling electroactivity with high-yield PHA biosynthesis, along with techno-economic assessments and environmental impact analyses, will be essential in transitioning from lab-scale systems to industrial applications.

In conclusion, BES-driven PHA production stands out as a promising avenue for the production of sustainable bioplastics using renewable electricity and waste-derived feedstocks. Continued interdisciplinary research and innovation will be key to realizing its full potential and establishing this technology as a viable alternative to conventional plastic manufacturing.

## Figures and Tables

**Figure 1 bioengineering-12-00616-f001:**
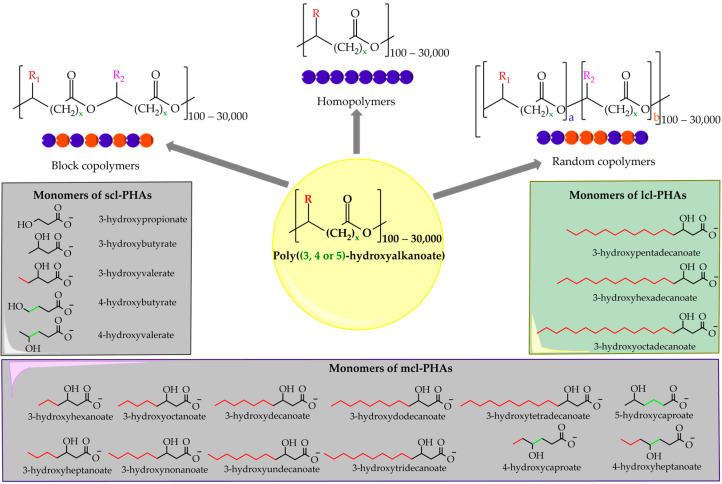
Structure and classification of polyhydroxyalkanoates (PHAs), highlighting their structural and compositional diversity. Modified figure from [22].

**Figure 2 bioengineering-12-00616-f002:**
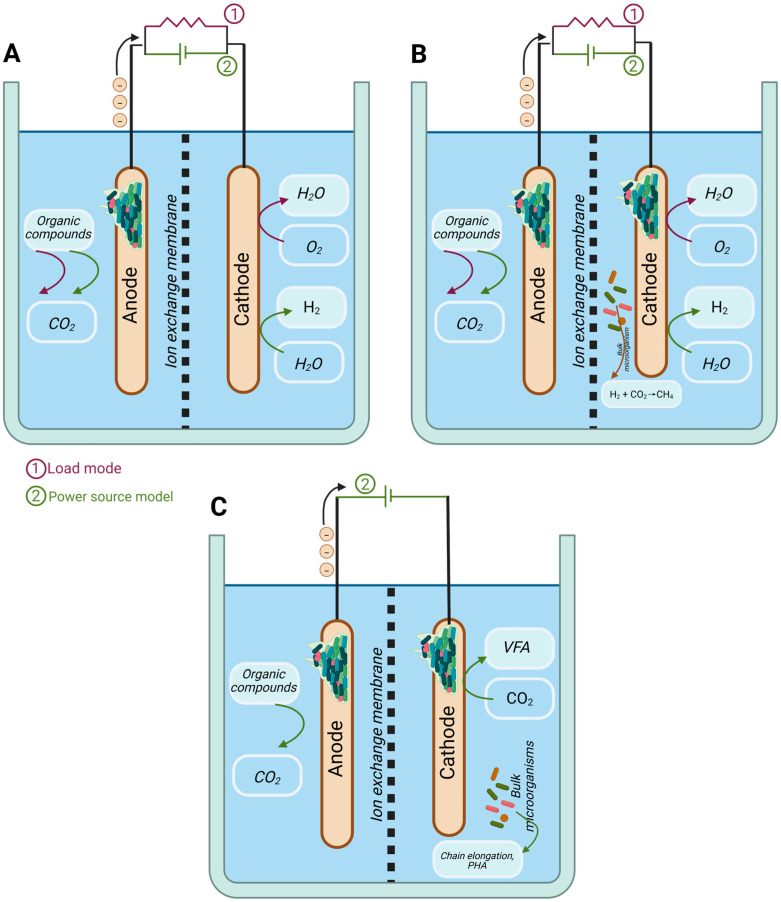
Evolution of bioelectrochemical system (BES) configurations for value-added product generation. The figure illustrates different configurations utilized in the evolution of bioelectrochemical systems. (**A**) displays both a microbial fuel cell (MFC) and a microbial electrolysis cell (MEC) configuration. In both, the anodic reaction is the same: the oxidation of organic compounds to CO_2_. However, they differ in the cathodic reduction reaction: in an MFC, the cathode typically results in the production of H_2_O (generating electricity), while, in an MEC, an external power source is applied to drive the reduction at the cathode, often resulting in H_2_ production. (**B**) presents an MFC and MEC configuration featuring two catalyzed electrodes, highlighting the first instance of electrosynthesis for methane generation from H_2_ and CO_2_ or CH_4_ at the cathode. (**C**) depicts an electrosynthesis (MES) reactor designed for the production of more complex organic molecules, such as volatile fatty acids (VFA) or PHA, from CO_2_. In the latter setup, the significance of microorganisms suspended in the bulk liquid for chain elongation and PHA production is emphasized.

**Table 1 bioengineering-12-00616-t001:** Microorganisms producing PHAs and exhibiting bioelectrochemical activity.

Domain	Microorganism	PHAs *	Bioelectrochemical Behavior	References
**Bacteria (Prokaryotes)**				
Burkholderiaceae	*Burkholderia cepacia*	scl- and mcl-PHAs	Electrotrophic	[5,51]
	*Cupriavidus metallidurans*	scl-PHAs	Electrotrophic	[5,52]
	*Cupriavidus necator*	scl-PHAs	Electrotrophic	[3,5]
Clostridiaceae	*Clostridium butyricum*	scl-PHAs	Electrotrophic	[5,53]
	*Clostridium pasteurianum*	scl-PHAs	Exoelectrogenic	[5,53]
Comamonadaceae	*Comamonas testosteroni*	scl-PHAs	Exoelectrogenic	[5,54]
Micrococcaceae	*Micrococcus luteus*	scl-PHAs	Electrotrophic	[5,55]
Propionibacteriaceae	*Propionibacterium* spp.	scl-PHAs	Electrotrophic	[5,56]
Pseudomonadaceae	*Pseudomonas aeruginosa*	mcl-PHAs	Exoelectrogenic	[5,57]
	*Pseudomonas alcaliphila*	mcl-PHAs	Exoelectrogenic	[5,58]
Rhodobacteraceae	*Rhodopseudomonas palustris*	scl-PHAs	Exoelectrogenic	[5,59]
**Cyanobacteria (Prokaryote)**				
Synechococcaceae	*Synechococcus elongatus*	PHB	Exoelectrogenic	[60,61]
Spirulinaceae	*Spirulina platensis*	PHB	Exoelectrogenic	[62,63,64]
**Algae (Eukaryotes)**				
Chlorellaceae	*Chlorella vulgaris*	PHB	Exoelectrogenic and Electrotrophic	[62,65,66]

* Note: scl-PHAs—short-chain-length PHAs; mcl-PHAs—medium-chain-length PHAs.

## Data Availability

No new data have been created.

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
