# Peer review of "A Review of Bioelectrochemical Strategies for Enhanced Polyhydroxyalkanoate Production"

_bioengineering, 2025, doi:10.3390/bioengineering12060616_

Round 1

Reviewer 1 Report

Comments and Suggestions for Authors

The manuscript titled “A Review of Bioelectrochemical Strategies for Enhanced Polyhydroxyalkanoate Production” mainly discusses the recent advancements in bioelectrochemical technologies for the production of polyhydroxyalkanoate (PHA). The authors clearly explained the potential of bioelectrochemical systems (BES) and the uses of BES to enhance the PHA production.  Overall, this review is well-structured, properly cited, and written concisely. I recommend a Minor revision of the manuscript. However, there are some minor modifications and clarification will be addressed before its final publication. Please revise the manuscript according to the following comments. 

Comments: 

Abstract: The Abstract was well constructed 

Keywords: Keywords are relevant to the topic.   

General comments: 

The introduction was written well.

Line 191: Bacillus megaterium should be in italics

Line 195: Alcaligenes latus should be in italics

Table 1: The abbreviations, such as scl and mcl-PHAs, should be explained in the footnotes of the table. 

The Eukarya should be changed to Eukaryotes, but in the same table, to list the bacterial species, it was mentioned as bacteria, not Prokaryotes, so the author should consider to use  Bacteria (Prokaryotes) and Algae (Eukaryotes) as a title. 

Line 284-293: The Author has mentioned the genetic engineering of the PHA producing pathway and its optimization, but hasn't included the names of the genes or pathway involved in the production of PHA. It would be better to show a few gene clusters of PHAs-producing bacteria and discuss more about the differences. Since this journal is Bioengineering, it would be more appropriate to provide this informations to the readers. 

Fig 2: The figure captions should have all the details mentied in the figure, it should be independent of the manuscript and shoul give detailed explanations of the figure. 

Line 363: Thermoanaerobacter kivui should be in italics

Language Usage: Language usage was fine.

Author Response

First of all, we would like to thank the reviewers and the editor for their valuable time and their dedication, which let us improve the manuscript. Reviewers’ comments have been entertained point by point below and the changes are highlighted in red color.

General comments: 

The introduction was written well.

Line 191: Bacillus megaterium should be in italics

Line 195: Alcaligenes latus should be in italics

Table 1: The abbreviations, such as scl and mcl-PHAs, should be explained in the footnotes of the table. 

Line 363: Thermoanaerobacter kivui should be in italics

The Eukarya should be changed to Eukaryotes, but in the same table, to list the bacterial species, it was mentioned as bacteria, not Prokaryotes, so the author should consider to use  Bacteria (Prokaryotes) and Algae (Eukaryotes) as a title. 

Response to general comments:  We apologise for these oversights and thank the reviewer for their comments. All italics in the text and in Table 1 have been corrected and highlighted in red for review.

Comment 1: Line 284-293: The Author has mentioned the genetic engineering of the PHA producing pathway and its optimization, but hasn't included the names of the genes or pathway involved in the production of PHA. It would be better to show a few gene clusters of PHAs-producing bacteria and discuss more about the differences. Since this journal is Bioengineering, it would be more appropriate to provide this informations to the readers. 

Response 1: We thank the reviewer for this comment. In response, we have revised the manuscript to include specific information on the genetic basis of PHA biosynthesis. We now describe key genes such as phaA, phaB, and phaC from the phaCAB operon in Cupriavidus necator and relevant gene clusters in Pseudomonas putida involved in mcl-PHA production. The differences in enzymatic pathways and regulatory mechanisms among these organisms are also briefly discussed, providing readers with a clearer understanding of the bioengineering strategies applied to enhance PHA synthesis.

Comment 2:  Fig 2: The figure captions should have all the details mentied in the figure, it should be independent of the manuscript and shoul give detailed explanations of the figure. 

Response 2: We appreciate your comment and have improved the caption for the figure as you suggested.

Reviewer 2 Report

Comments and Suggestions for Authors

This somewhat brief review covers the emerging field of bioelectrochemical synthesis of biologically compatible and biodegradable polymers. The field is very interesting and important and a review is needed to summarize the important recent literature. In this context, the manuscript is highly relevant.

The manuscript is written well and organized nicely. A few examples of successful bioelectrochemical synthesis procedures are provided and material yields are listed. However, I have a few significant complaints about this review.

First, the abstract promises “a focus on the role of electrode potentials and microbial electron transfer mechanisms in improving polymer yield” (line 27). No such information or discussion is presented from what I can read. Table 1 lists examples of organisms but no electrode potentials or yields, or any other quantitative results, are listed in this table. Section 5 provides yields with the main text but it is difficult to connect the examples in the text with the examples in Table 1, and especially difficult to follow and connect the electron transfer mechanisms with improved yields. The authors need to include applied voltages and other basic electrochemical parameters in their literature analysis.

Second, Figure 2 does not really provide much information. This figure is modified from a literature figure (ref 134). The original figure in the literature is much better, and Figure 2 in the present work is essentially redrawn but without as many labels. Anyway this figure is pretty much a picture of any electrochemical reactor that one would find in a freshman chemistry or chemical engineering textbook. What is the stuff coated onto the electrodes? What is happening on the abiotic electrodes? How is the synthesized material collected from this reactor? The text (section 3.2) refers to different reactor configurations, but the reader only has the oversimplified Figure 2 for reference. The authors need to find a way to better describe the reactors with information that would be useful and informative.

Finally, section 4 provides hints about electrode potentials and improved materials production but again provides no quantitative summaries from the literature. The authors need to include details from the literature (voltages, materials, etc) from diverse studies to connect the reactor designs to the reasons for improved yields.

The review seems interesting but is a little brief and incomplete as it does not present details such as voltages and diverse reactor designs from the literature sources reviewed. The authors can and should include more information in Table 1 and Figure 2 (and perhaps new figures) to better inform the readers about how the electrochemical parameters connect to the polymer production.

Author Response

First of all, we would like to thank the reviewers and the editor for their valuable time and their dedication, which let us improve the manuscript. Reviewers’ comments have been entertained point by point below and the changes are highlighted in red color.

Comment 1: First, the abstract promises “a focus on the role of electrode potentials and microbial electron transfer mechanisms in improving polymer yield” (line 27). No such information or discussion is presented from what I can read. Table 1 lists examples of organisms but no electrode potentials or yields, or any other quantitative results, are listed in this table. Section 5 provides yields with the main text but it is difficult to connect the examples in the text with the examples in Table 1, and especially difficult to follow and connect the electron transfer mechanisms with improved yields. The authors need to include applied voltages and other basic electrochemical parameters in their literature analysis.

Response 1: Regarding Comment 1, we fully agree with your assessment. We acknowledge that the initial submission lacked sufficient detail and discussion on the role of electrode potentials and microbial electron transfer mechanisms in improving polymer yield, as promised in the abstract. In response to your feedback, we have thoroughly revised the literature analysis in Section 4 and 5. We have now incorporated specific information regarding applied voltages and other basic electrochemical parameters (such as electrode materials and relevant operational conditions) from key studies. This expanded discussion aims to provide a clearer connection between the electrochemical conditions, microbial electron transfer mechanisms, and their impact on polymer yield. All these specific additions and modifications within the manuscript have been highlighted in red for your convenience in reviewing the changes.

Comment 2: Second, Figure 2 does not really provide much information. This figure is modified from a literature figure (ref 134). The original figure in the literature is much better, and Figure 2 in the present work is essentially redrawn but without as many labels. Anyway this figure is pretty much a picture of any electrochemical reactor that one would find in a freshman chemistry or chemical engineering textbook. What is the stuff coated onto the electrodes? What is happening on the abiotic electrodes? How is the synthesized material collected from this reactor? The text (section 3.2) refers to different reactor configurations, but the reader only has the oversimplified Figure 2 for reference. The authors need to find a way to better describe the reactors with information that would be useful and informative.

Response 2: In response to your suggestion, we have completely revised Figure 2 to better serve its purpose within the manuscript. The new Figure 2 is now composed of distinct panels (A, B, and C) specifically designed to illustrate the evolution of different bioelectrochemical system configurations across various stages of development. Our aim with this revised figure is to provide a much clearer conceptual overview of how these reactor setups have progressed, from basic MFC/MEC designs to more complex electrosynthesis systems for specific products.

We believe that this modification allows for a clearer understanding of the distinct configurations and their progression, making the information much more accessible and informative for the reader. Specific details regarding electrode materials, applied potentials, and the collection of synthesized materials, which were your previous concern, are now comprehensively discussed in the relevant sections of the main text, as addressed in our response to Comment 1.

Comment 3: Finally, section 4 provides hints about electrode potentials and improved materials production but again provides no quantitative summaries from the literature. The authors need to include details from the literature (voltages, materials, etc) from diverse studies to connect the reactor designs to the reasons for improved yields.

Response 3: Thank you for your final comment regarding Section 4. We appreciate your suggestion to include more quantitative details from the literature. In response to your feedback, we have thoroughly revised Section 4. We have now incorporated the requested quantitative information, including specific voltages and electrode materials from diverse studies, to better connect the reactor designs with the reasons for improved yields. These additions aim to provide a more comprehensive and data-driven discussion in this section. As with the previous changes, all these specific modifications have been highlighted in red in the manuscript for ease of identification.

Round 2

Reviewer 2 Report

Comments and Suggestions for Authors

The authors have addressed my comments and have included a revised figure (Fig. 2). The manuscript is now acceptable for publication.